# Battery-Powered Portable Rotary Real-Time Fluorescent qPCR with Low Energy Consumption, Low Cost, and High Throughput

**DOI:** 10.3390/bios10050049

**Published:** 2020-05-08

**Authors:** Limin He, Benliang Sang, Wenming Wu

**Affiliations:** 1State Key Laboratory of Applied Optics, Changchun Institute of Optics, Fine Mechanics and Physics (CIOMP), Chinese Academy of Sciences, Changchun 130033, China; helimi18@mails.ucas.ac.cn (L.H.); sangbenliang18@mails.ucas.edu.cn (B.S.); 2University of Chinese Academy of Sciences (UCAS), Beijing 100049, China

**Keywords:** real-time PCR, portable, rotary, low cost, low energy consumption, high throughput

## Abstract

The traditional qPCR instrument is bulky, expensive, and inconvenient to carry, so we report a portable rotary real-time fluorescent PCR (polymerase chain reaction) that completes the PCR amplification of DNA in the field, and the reaction can be observed in real-time. Through the analysis of a target gene, namely pGEM-3Zf (+), the gradient amplification and melting curves are compared to commercial devices. The results confirm the stability of our device. This is the first use of a mechanical rotary structure to achieve gradient amplification curves and melting curves comparable to commercial instruments. The average power consumption of our system is about 7.6 W, which is the lowest energy consumption for real-time fluorescence quantification in shunting PCR and enables the use of our device in the field thanks to its self-contained power supply based on a lithium battery. In addition, all of the equipment costs only about 710 dollars, which is far lower than the cost of a commercial PCR instrument because the control system through mechanical displacement replaces the traditional TEC (thermoelectric cooler) temperature control. Moreover, the equipment has a low technical barrier, which can suit the needs of non-professional settings, with strong repeatability.

## 1. Introduction

Real-time PCR (polymerase chain reaction) amplification was introduced in 1995 [1]. PCR is a widespread nucleic acid analysis method with high precision and high sensitivity. However, a traditional commercial PCR instrument is bulky and expensive and requires a laboratory setting, which is not suitable for field operation and prevents its application to disease diagnosis in remote areas. A quantitative PCR instrument uses fluorescent chemical detection to monitor the DNA amplification in real-time [2,3,4]. The fluorescent dyes added will combine to double-stranded DNA to emit fluorescence and allow characterizing the PCR cycling [5]. Since real-time fluorescence quantitative PCR was introduced, it evolved from a qualitative to a quantitative approach. With the advantages of strong specificity, high sensitivity, and simple operation, and because it eliminates the need for post-processing the reaction products by, for instance, electrophoresis and hybridization, quantitative PCR has played an important role in scientific research and has been widely used for clinical diagnosis. It has become the standard for the detection of DNA [6]. In clinical diagnosis, it is used for early diagnosis and treatment tracking, and in genetic research. For example, it had a significant influence on the detection of avian influenza, hepatitis B, AIDS (acquired immunodeficiency syndrome) and other viruses [7,8,9,10,11,12]. In scientific research, PCR is used for basic research such as species identification, gene isolation, cloning, and nucleic acid sequence analysis. It is recognized as the gold standard for nucleic acid analysis [13,14,15,16].

After 20 years of development, the temperature cycle for PCR amplification has evolved considerably. Chamber-based stationary PCR, continuous-flow PCR, and shunting PCR are the three main approaches for PCR devices [17].

The chamber-based stationary PCR is the most common technology for portable PCR. It only has one heater connected directly to the chip. The temperature cycling is accomplished by controlling the heater’s temperature. The heating devices in chamber-based stationary PCR are film heaters [18,19], silicon heaters [20], and so on. The most widespread approach is to control thermoelectric components (TECs) to achieve temperature cycling [21,22]. A real-time fluorescence quantitative PCR instrument with a TEC uses an h-bridge circuit to control the forward and reverse currents in the semi-conductors to cycle the temperature. It is a mature technology, and many commercial instruments have been produced. Representative companies that Bio-Rad (Hercules, CA, USA), ABI (Applied Biosystems, New York, NY, USA), Biosynex (Alsace, France), ThermoFisher (Waltham, MA, USA) built a battery-powered, real-time fluorescent PCR instrument using TEC heating and cooling [23]. The TEC is simple and easy to integrate but is time-consuming and has a high power consumption. Typically, 30 cycles consume a power of 6 Ah [23].

The main goals of all of the approaches are to reduce the amount of expensive reagents used and achieve a simpler structure to produce better PCR instruments [24]. The advantage of a chamber-based stationary PCR is that the products can be easily collected for further analysis by electrophoresis, for instance. This kind of equipment is mature and has a stable performance, but a majority of the equipment from the above companies is large and has a high energy consumption. This prevents popularizing the technology in remote areas.

Other methods for cycling the temperature have been proposed to meet the needs of a rapid, real-time, in-field analysis. One of them is continuous-flow PCR, where several heaters are used at a different temperature. The temperature cycling is carried out by moving the chip between the heaters. In this system, pipes are fixed on the heaters, and the reagents flow through the pipes and pass through the different heaters to achieve temperature cycling. Our team recently designed a PCR technology based on continuous flow and reduced the energy consumption to less than 10 watts [25].

The other approach is shunting PCR, where a series of chamber-based stationary and continuous-flow PCRs are used. It couples multiple heaters with the chip by mechanically moving the chip between different temperature regions and completing a temperature cycle. This method has several advantages, including rapid circulation, easy collection of the products, a simple structure, a low technical barrier, and high repeatability. Salman et al. designed a shunting PCR device for the rapid detection of bacteria [17]. It has a short reaction time, but it is still not used in real-time PCR equipment because it is difficult to perform an electrophoresis post-amplification. Jung et al. proposed a new platform for an ultra-fast PCR system used for high sensitivity and rapid identification of the influenza virus RNA [26]. A drawback of the system is that it cannot be battery-operated and cannot be isolated from a laboratory. Furthermore, the amplification products need to be moved to special instruments for testing that cannot also detect the products.

A series of important breakthroughs have been made in continuous-flow PCR and mechanical-displacement PCR. Continuous-flow PCR can easily shorten the heating and cooling time with a simpler thermal cycle [27]. Shunting PCR combines the advantages of chamber-based stationary PCR and continuous-flow PCR [17]. Shunting PCR has a simpler structure for rapid heating and cooling and amplification products can be easily collected for their follow-up analysis.

In Table 1, we compare several representative portable PCR instruments. The existing portable PCR instrument does not have the characteristics of low cost and low energy consumption and the function of real-time quantitative fluorescence. We propose a portable rotary real-time fluorescent PCR with low energy consumption, low cost, a low technical barrier, and high throughput. The size of the system is about 180 × 120 × 250 mm. It includes a high-throughput chip, a thermal cycling system, and a fluorescence-detection system. The chip is an inexpensive, disposable, multi-chamber chip made of polydimethylsiloxane (PDMS) and a glass chip that can amplify multiple reactions at the same time, including a negative control. The automatic control system has been developed and applied to simultaneous timing and coupling, including mechanical rotation, temperature cycle, excitation of light, and image acquisition. In the thermal cycling system, a constant-temperature heat source is rotated by a steering gear to control the temperature of the chip. The fluorescence-detection system collects the fluorescence signal of the chip in the last 3 s of the extension stage of each cycle. The reaction of the reagent is observed in real-time by monitoring the fluorescence intensity in the chip cycle by cycle to realize the drawing of an amplification curve. For the step-by-step heating step of the low temperature heater, the drawing of the melting point curve is realized and the automatic image-processing software is developed and applied to the automatic analysis of the amplification curve and the melting curve. This real-time fluorescence PCR system not only achieves the function of a commercial real-time fluorescence quantitative PCR instrument but also has a low energy consumption, a very low cost, a low technical barrier, and high throughput, so that it can play a significant role in real-time diagnosis.

## 2. Materials and Methods

The core of our real-time fluorescence quantitative PCR device has two aspects: the temperature cycle and the real-time fluorescence detection. The temperature cycle is the key for a successful real-time fluorescence quantitative PCR. The DNA has to undergo three processes of denaturation, annealing, and extension in an environment with a suitable temperature. The amplification of the DNA is contained in the fluorescence signal as collected by the detection system for further processing and analysis (Figure 1a).

### 2.1. Thermal Cycling System

#### 2.1.1. Amplified Heat Cycle System

The thermal cycling system is the heart of a PCR system. It provides the appropriate amplification temperature to the reagents in the chip, and the system must automatically complete the temperature change and cycle. In this study, mechanical rotary thermal coupling is used to define a temperature cycle. The structure of the cycle mainly has two parts, namely the steering gear rotating system and the constant-temperature heat source, as shown in Figure 1b. The steering gear rotation system consists of a steering gear (LX-16A, Feilai, Jiangsu, China), a chip support platform with a bottom made of a copper bearing chip, and a steering gear cage. The constant-temperature heat source consists of 60 °C and 95 °C sources. It is composed of a PTC (positive temperature coefficient) constant-temperature heating sheet (Xidebao, Shenzhen, China), a thermostat (TCM-M207, Yexian, Sichuan, China), and a heat-source cage.

The mechanical rotation system is a steering gear that rotates around the intermediate axis. The 60 °C heat source and the 95 °C heat source are fixed at both ends of the heat-source cage and rotate with the steering gear. In the initial stage of the reaction, the chip is heated on the constant-temperature heat source at 95 °C for 30 s to complete the high-temperature denaturation of the DNA. In the cooling stage, the steering gear drives the heating source to rotate 180° counterclockwise to cool the chip rapidly to 60 °C for 15 s. Then, the steering gear is cooled to 60 °C in ambient air for 12 s. Next, the chip is maintained at a constant temperature of 60 °C for 30 s to complete the annealing and extension of the DNA. At this point, the chip completes one cycle of DNA replication. The steering gear will then rotate 180° clockwise to place the 95 °C heat source under the chip for the next denaturation cycle. The heat source driven by the steering gear repeatedly heats and cools the chip at different temperatures to complete the PCR amplification. Figure 1c–e shows the temperature cycling diagram.

It takes 93 s to complete a temperature cycle, and it takes 62 min for a PCR reaction with 40 cycles. The chip support platform designed in this work can be fine-tuned by adjusting the position of the chip support plate with bolts and nuts so that the chip maintains a good connection directly above the constant-temperature heat source. To address the gap between the heat source and the chip support frame, we added a layer of thermal conductive silicone grease on the heating sheet (HT-WT160, Ketian Microelectronics Co., Ltd., Shenzhen, China) to improve the overall thermal conductivity.

#### 2.1.2. Melting Heat Cycling System

Because the dyes used have no specificity and can combine with other products to emit fluorescence, other products such as primer dimers can be produced during the amplification. After PCR amplification, a melting curve has to be recorded to test whether the products are clean and pure.

In the thermal cycling system, the heat source at 60 °C stays under the chip after the last cycle. The melting heat cycle controls the increase of the temperature from 60 °C to 95 °C. When the temperature of the heating sheet is increased by 1 °C for 10 s, the fluorescence signal collected by the micro-camera is analyzed by a computer.

### 2.2. Optical Detection System

The prolonged irradiation of the light source leads to the deterioration and discoloration of the dye. This means that the fluorescence detection system can only collect the fluorescence signal at set intervals. The fluorescence detection system is composed of an excitation light source and a signal-acquisition device. The excitation light source is composed of a light source (T850AC1670GD-P488, Anford, Shenzhen, China), an optical filter (Anford, Shenzhen, China), a relay (LCUS-1, Telesky, Shenzhen, China), and a control circuit. The signal acquisition module is composed of a miniature camera (MV-CB120-10UC-C, Hikvision, Hangzhou, China) with an image resolution of 4032 × 3036, a spectral filter (Yizheng laser, Beijing, China), and a signal-transmission module.

The fluorescence collection is activated during the last 3 s of the extension step. The computer controls the serial port of the relay to close it and disconnect it at a specific times. When the relay is closed, the light illuminates the microfluidic chip. After 1 s, the computer orders the miniature camera to collect the fluorescence and uploads the fluorescence information for each cycle to the computer. After 3 s, the relay circuit is disconnected to collect the fluorescence in intervals. A fluorescence curve can be obtained through the analysis of the fluorescence intensity in each recorded photograph. Figure 2 shows a workflow diagram of the system.

### 2.3. Microfluidic Chip

In recent years, micro-droplet approaches have greatly reduced the thermal inertia of the original microfluidic systems. They use two incompatible fluids to form droplets or emulsions in the chamber, and these emulsions define the basic units for PCR amplification.

The chip is made by mixing PDMS and a curing agent at a ratio of 10:1 and placing the mixture in a slight vacuum for 30 min to remove the bubbles. After curing, the PDMS piece is cut to an appropriate size. The chamber is made with a puncher, and a 0.1 mm glass sheet is bonded by plasma technology to define a chip. The size of the chip is 20 × 20 mm. A total of 2 μL of reagent is placed in the chamber of the chip chamber and covered with mineral oil to prevent water evaporation.

Such a chip can define multiple reaction chambers to achieve a higher throughput. It can be used to react many reagents simultaneously and include a negative control. The chip has the advantages of simple manufacturing, low cost, and high throughput, which makes the cost of an experiment lower and the efficiency higher.

### 2.4. Reagents

The PCR reagents consisted of 10 μL of PreMix Taq, 6 μL of DD water (double distilled H_2_O), 2 μL of polymerase chain reaction template, 1 μL of EvaGreen dye, and 1 μL of forward and reverse primers. The template was plasmid DNA pGEM-3Zf (+). The forward primer sequence was CCAGTCGGGAAACCTGTCGTGCC, and the reverse primer sequence was GTGAGCGAGGAAGCGGAAGAGCG.

### 2.5. Power Management

Rechargeable lithium batteries (12 V, 1800 mAh, 59 × 35 × 15 mm) were used for the constant-temperature heat sources, the steering gear, and the excitation light source (Figure 3a). The current change measured at 12 V is shown in Figure 3b. The average power per cycle was 7.6 W, and 40 cycles took 62 min to complete, with a total power consumption of around 0.65 Ah.

## 3. Results and Discussion

### 3.1. Analysis of the Temperature Curve

In the PCR amplification stage, the cooling rate is one of the main factors affecting the overall reaction time. We proposed three different cooling modes and compared their performance differences in the temperature cycle. In mode 1 (Figure 4a, curve 1), the chip is first heated to 95 °C and then the temperature is reduced from 95 °C to 60 °C in air; the steering gear maintains the constant-temperature source at 60 °C at the bottom of the chip. In mode 2 (Figure 4a, curve 2), the chip is cooled and the temperature is maintained directly by the 60 °C heat source. Figure 4a shows that the temperature drop in mode 1 is relatively uniform, with a good plateau. The temperature decreases faster in curve 2 than in curve 1 between 95 °C and 68 °C, but is slower between 68 °C and 60 °C. Mode 3 (Figure 4a, curve 3) combines the advantages of the other two modes by using a constant-temperature heat source at 60 °C when the temperature is reduced from 95 °C and cooling the chip in air to reduce the temperature from 68 °C to 60 °C. Figure 4a shows that the temperature cycling of curve 3 is the fastest. The cooling characteristics of the various modes are analyzed in Table 2.

We used 95 °C for 10 s for the denaturation of the template DNA and 60 °C for 30 s for the template annealing and extension. Figure 4b shows that the time can be adjusted. Curves 1, 2, and 3 are the temperature cycling at 95 °C and 60 °C for the different times tested. Figure 4c shows the melting temperature cycle curve obtained by using a Python script to read the port data and control the temperature change.

### 3.2. Analysis of the Gradient Template Amplification Curve

We used a gradient concentration template to amplify 40 cycles simultaneously on both our portable PCR system and commercial qPCR instruments. Figure 5a shows that we selected the fluorescence intensity values collected from three experiments to plot the amplification curve with its standard deviation. Figure 5b shows the amplification curve of the commercial PCR instrument. The commercial PCR instrument has additional software to process the resulting data, and the curve is smoother. The number of cycles corresponding to the intersection of the fluorescence threshold and the amplification curve is the Ct value. We used a fluorescence threshold of 4, and the Ct was 15.9, 19.0, 22.9, and 25.5. The Ct on the commercial PCR instrument was 13.2, 16.5, 20.5, and 23.8. Figure 5c–f shows the real-time fluorescence signal and the reaction for each concentration from the beginning to the end of the cycle.

### 3.3. Melting Curve Analysis

At the end of the amplification reaction, a melting curve can be drawn by gradually increasing the temperature and monitoring the fluorescence brightness of each temperature stage at the same time. Different DNA double helix structures have different temperatures when the melting rate is the fastest and, therefore, different characteristic peaks (Tm). Whether or not the characteristic peak is at the same temperature, it can be used to determine whether there is only one product from the DNA amplification. Since false positives occur during amplification, we added the ability to record a melting curve to test whether there are other products like primer dimers. Figure 6a shows a melting curve from the portable PCR instrument, and Figure 6b shows the one on a commercial PCR instrument. The characteristic peak appears at the same temperature, indicating that the product is relatively pure.

## 4. Conclusions

We designed and tested a shunting real-time PCR device with a low energy consumption. The average power consumed during the thermal cycling is 7.6 W. The PCR system has a simple structure, a low price (about US $710), and a low technical barrier in that we do not use customized devices and all of the devices can be obtained easily, so that it has good repeatability. The disposable chip used in the system is simple and inexpensive to fabricate and has a high throughput, which reduces the cost of an experiment. We proved the stability of our device to amplify the pGEM template and test the PCR products. Our portable real-time PCR device has a low energy consumption, a low cost, and a low technical barrier, which gives it the potential to be a popular device for disease diagnosis in remote areas.

## Figures and Tables

**Figure 1 biosensors-10-00049-f001:**
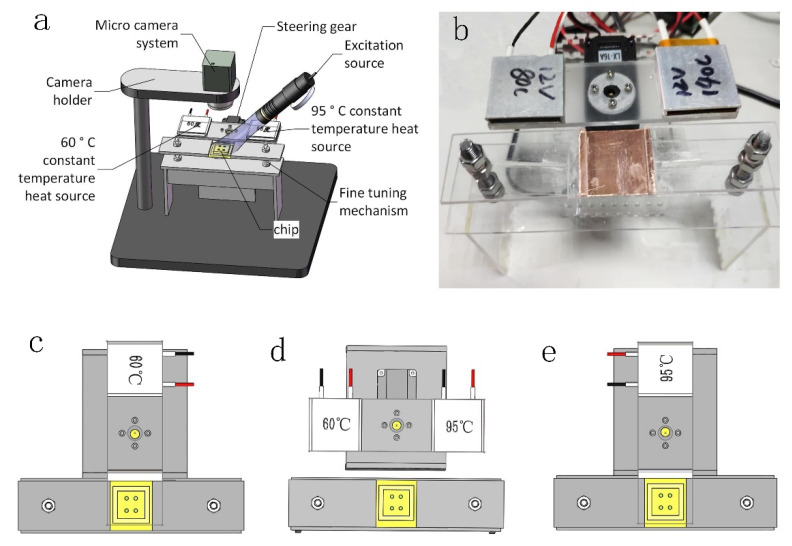
(**a**) Structural drawing of the battery-operated portable qPCR system. (**b**) Close-up of the chip holder. (**c**) Configuration for the (**c**) 95 °C heating, (**d**) air cooling, and (**e**) 60 °C cooling with a constant temperature.

**Figure 2 biosensors-10-00049-f002:**
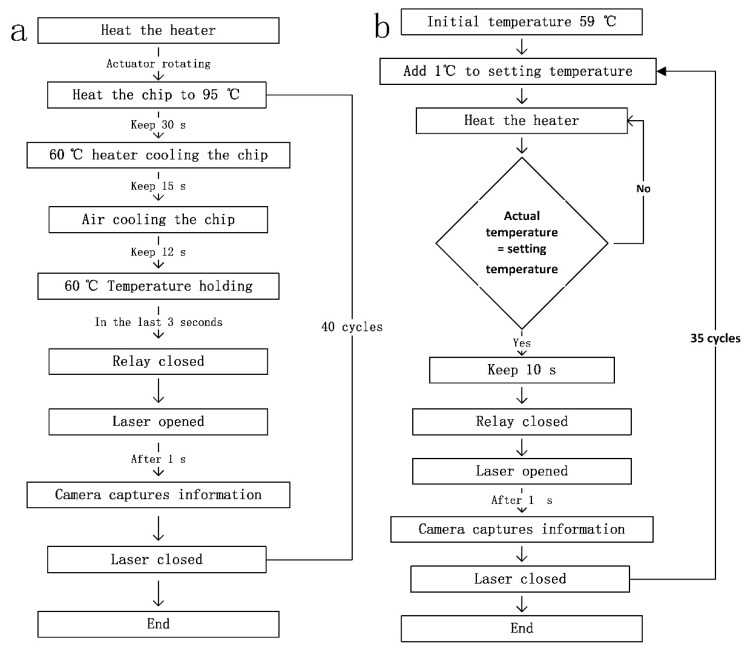
Diagram in mode 3 for the (**a**) amplification workflow and (**b**) the melting curve.

**Figure 3 biosensors-10-00049-f003:**
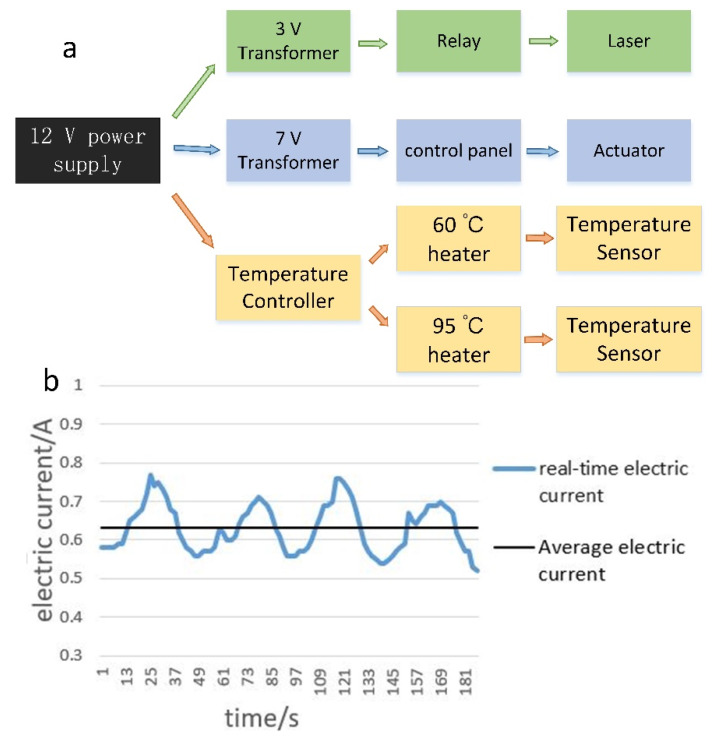
(**a**) Power supply shunt diagram. (**b**) Current variation at 12 V over two cycles.

**Figure 4 biosensors-10-00049-f004:**
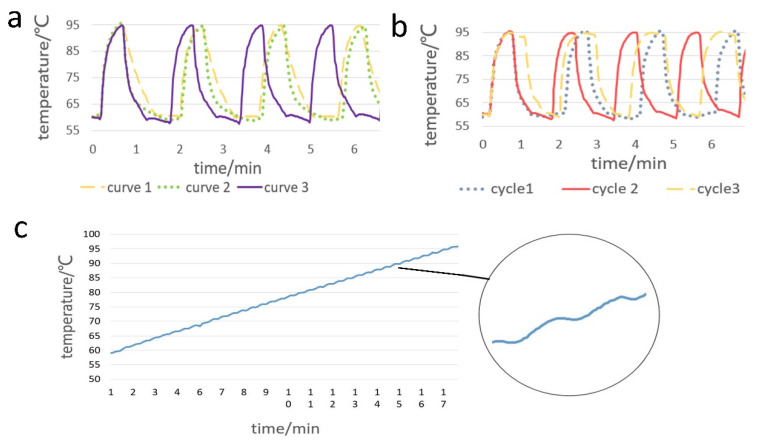
(**a**) Curve 1 is the temperature curve when the chip is cooled to about 60 °C in air. Curve 2 corresponds to the curve when the chip is cooled directly by a 60 °C heat source. Curve 3 corresponds to first cooling with a heat source at 60 °C and then in air. (**b**) Temperature cycling for different holding times at 95 °C and at 60 °C. (**c**) Melting curve temperature profile when the temperature gradually increases from 60 °C to 95 °C with a 1 °C interval. The enlarged circle is a partially enlarged view.

**Figure 5 biosensors-10-00049-f005:**
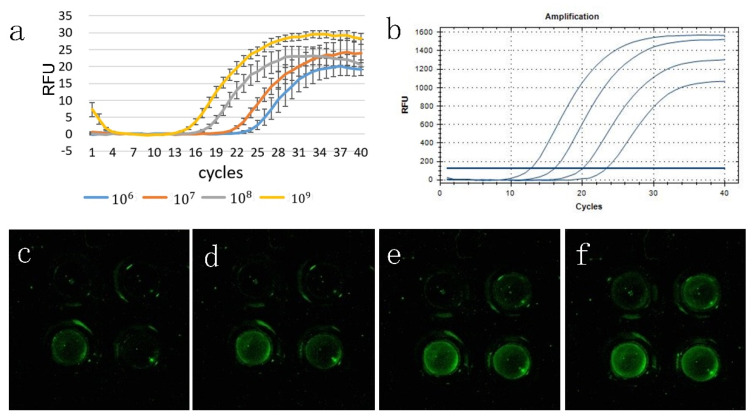
Gradient amplification curve on (**a**) the portable PCR system and the (**b**) commercial PCR instrument. (**c**–**f**) Fluorescence signal for the real-time fluorescence image acquired by the detection system when the number of cycles is 17, 20, 24, and 27, fluorescence changes of different initial concentrations of DNA that can be observed. The initial concentration is 10^6^ in the lower left corner, 10^7^ in the lower right corner, 10^8^ in the upper right corner, and 10^9^ in the upper left corner.

**Figure 6 biosensors-10-00049-f006:**
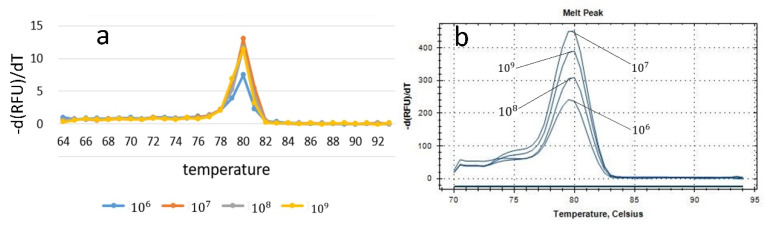
Fore gradient concentrations melting curve on (**a**) the portable PCR device and (**b**) the commercial instrument.

**Table 1 biosensors-10-00049-t001:** The comparison of portable PCR instruments.

Reference	Journal/Years	Temperature Control Mode	Energy Consumption	Gradient Amplification Curve	Melting Curve
[23]	IEEE Explore 2016	TEC	6 ah (30 cycles)	provided	provided
[25]	Sensors 2019	Constant temperature heater	6 W	Not provided	Not provided
[17]	Talanta 2020	Translational motion	Not provided	Not provided	Not provided
[26]	Lab chip 2012	rotation	Not provided	Not provided	Not provided
This study	2020	rotation	7.6 W	provided	provided

**Table 2 biosensors-10-00049-t002:** Temperature cycling characteristics of the different cooling modes.

Cooling Mode	Overall Cooling Time	Overall Cooling Rate	95–68 °C Cooling Rate	68–60 °C Cooling Rate
mode 1	45 s	0.77 °C/s	0.9 °C/s	**0.53 °C/s** ^1^
mode 2	47 s	0.74 °C/s	**2.25 °C/s**	0.23 °C/s
mode 3	28 s	**1.25 °C** **/** **s**	**2.25 °C** **/** **s**	**0.53 °C** **/** **s**

^1^ Bold fonts are used to indicate that the temperature drops faster.

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
