# Peer review of "Battery-Powered Portable Rotary Real-Time Fluorescent qPCR with Low Energy Consumption, Low Cost, and High Throughput"

_biosensors, 2020, doi:10.3390/bios10050049_

Round 1

Reviewer 1 Report

In this paper, the author has completed a device which can be used for nucleic acid detection in remote areas. However, there are still several problems to be clarified and improved.

  1. In the introduction part, because the author's article is a portable instrument, why not explain the reasons for choosing RTPCR. Isn't digital amplification more suitable for portable instruments? Because there are many digital amplification instruments that can be portable. At the same time, the innovation of this article should be highlighted through the comparison with other literature in this part, but through the comparison of table readers cannot get too much information.
  2. Whether the author can provide the photos of the whole system and the chip, so that the reader can understand the work of the system more clearly. At the same time, the author should also introduce the following parameters, such as the size of the whole system, chip size, image resolution. If these parameters are not mentioned, this article is more like a popular science article. Readers can also judge from these parameters whether it belongs to portable instrument and whether your method is more novel and excellent.
  3. It is mentioned in the article that the computer script is used to control the acquisition of fluorescence image and data analysis, but it is no different from the laboratory instrument. The author should consider introducing the upper computer of the instrument
  4. There are still many details to be dealt with in the article, such as the situation of two words being linked together, the situation of language incoherence (“In Table1, we compared several representative portable PCR. It can be seen that the existing portable PCR cannot take into account of the characteristics of low cost, low energy consumption and real-time PCR), and the situation of Chinese in the picture (Figure 1)
  5. P4,“In the cooling stage, the steering gear drives the chip to rotate 180°counterclockwise to cool the chip rapidly to 60 ℃ for 15s”. The steering gear drives the chip or the heating source?
  6. When heating or cooling, is it necessary to ensure that the thermal conductive silicone grease and the chip are in close contact? If yes, does the constant rotation cause friction to cause the silicone grease or chips to wear and create voids? I think maybe there is a need to repeat the experiment to prove the reliability.
  7. P5,“In the thermal cycling system, the heat source at 60℃ stays above the chip after the last cycle.” The heat source stays above the chip or under the chip?

Author Response

Dear reviewers

Thank you for your letter dated April 23. We thank the reviewers for the time and effort that they have put into reviewing the previous version of the manuscript. The suggestions have enabled us to improve our work. Based on the instructions provided in your letter, we uploaded the file of the revised manuscript.

Appended to this letter is our point-by-point response to the comments raised by the reviewers. Our responses are given directly afterward in a different color (red).

We would like also to thank you for allowing us to resubmit a revised copy of the manuscript.

We hope that the revised manuscript is accepted for publication in the Journal of Biosensors.

Sincerely
Wenming Wu

Reviewer 2 Report

The manuscript is well-written and would be considered worthy of accept if the points shown below were corrected.

Throughout the entire sentence, there should be a space before and/or after numbers and parentheses. For example, “1995[1]” should be “1995 [1]” in L28, and “Table1” should be “table 1” in L87, “TELESKYChina” in L164, and so on. Again, the entire manuscript needs to be carried out by the authors.

L14: “PGEM-3ZF (+)” should be “pGEM-3Zf(+)”

After L129: “°C” font is different from that in L124

L143 & 171: “seconds” should be “s”

L190: “2” in the “H2O” should be a subscript character

L192: “PGEM-32F (+)” should be “pGEM-3Zf(+)”

L206, 208, 212:”Mode” should be “mode”

Paragraph 3.2.: Does the fact that the amplification cycle is slower than the commercial product imply poor amplification sensitivity?

L263: “PGEM” should be “pGEM”

Fig. 1: You should include a picture of the actual machine.

Fig. 2: The width of the box should be matched.

Fig. 2a: In the fifth box, “Temperature” is unnecessary.

This workflow is the mode 3 version? If that, indicate in the legend.

Fig. 3 & 4: cannot read the text because of the low resolution.

Fig. 3b: It would be better for the reader if a graph showed how many seconds to what process it was.

Fig. 4c: Need an explanation or annotation of what the enlarged circle indicates

Fig. 5 & 6: power number (6 of “10^6”) should be indicated by a superscript. And which line is from which initial DNA in (b)?

Fig. 5c–f: Indicate which well is from which initial DNA.

Table 1: Reference numbers in table 1 are off by one.

References: 1: unnecessary “1.” is inserted in the beginning.

Author Response

(The authors gave the same response as above.)

Round 2

Reviewer 1 Report

The authors have addressed all my concerns and therefore I have no further questions to this manuscript.